# Pre-pandemic geographic access to hospital-based telehealth for cancer care in the United States

**David I. Shalowitz** [1,2] *, **Peiyin Hung** [3], **Whitney E. Zahnd** [4], **Jan Eberth** [3,5]

**1** Department of Obstetrics and Gynecology, Section on Gynecologic Oncology, Wake Forest University School of Medicine, Winston-Salem, North Carolina, United States of America, **2** Department of Implementation Sciences, Wake Forest University School of Medicine, Winston-Salem, North Carolina, United States of America, **3** Rural & Minority Health Research Center, Arnold School of Public Health, University of South Carolina, Columbia, South Carolina, United States of America, **4** Department of Health Management and Policy, University of Iowa College of Public Health, Iowa City, Iowa, United States of America, **5** Department of Health Management and Policy, Dornsife School of Public Health, Drexel University, Philadelphia, Pennsylvania, United States of America

\* david.shalowitz@gmail.com

## Abstract

### Importance

Little is known about US hospitals' capacity to ensure equitable provision of cancer care through telehealth.

### Objective

To conduct a national analysis of hospitals' provision of telehealth and oncologic services prior to the SARS-CoV-2 pandemic, along with geographic and sociodemographic correlates of access.

### Design, setting, and participants

Retrospective cross-sectional analysis with Geographic Information Systems mapping of 1) 2019 American Hospital Association (AHA) Annual Hospital Survey and IT Supplement, 2) 2013 Urban Influence Codes (UIC) from the United States Department of Agriculture, 3) 2018 Area Health Resources Files from the Health Services and Resources Administration (HRSA).

### Interventions

Hospitals were categorized by telehealth and oncology services availability. Counties were classified as low-, moderate-, or high-access based on availability of hospital-based oncology and telehealth within their boundaries.

### Main outcomes and measures

Geospatial mapping of access to hospital-based telehealth for cancer care. Generalized logistic mixed effects models identified associations between sociodemographic factors and county- and hospital-level access to telehealth and oncology care.

**Editor:** Sonu Bhaskar, Global Health Neurology Lab / NSW Brain Clot Bank, NSW Health Pathology / Liverpool Hospital and South West Sydney Local Health District / Neurovascular Imaging Lab, Clinical Sciences Stream, Ingham Institute, AUSTRALIA

**Data Availability Statement:** The data underlying the results presented in the study are available from the American Hospital Association (https://www.ahadata.com/), the US Department of Agriculture (https://www.usda.gov/), the US Census https://www.census.gov/programs-surveys/geography/data/interactive-maps.html), and the Health Resources and Services Administration (https://data.hrsa.gov/). The authors did not have any special access privileges that others would not have. Datasets are accessible by following instructions on the above URLs.

**Funding:** Dr. Shalowitz received funding via the Wake Forest Comprehensive Cancer Center NCI cancer support grant (P30CA012197), and has received consulting fees from Nimble Co, LLC (purview.net) unrelated to the current work. the funders of this work had no role in study design, data collection and analysis, decision to publish, or preparation of the manuscript.

**Competing interests:** Funding and sources of competing interests do not alter our adherence to all PLOS ONE policies on sharing data and materials.

## Results

2,054 out of 4,540 hospitals (45.2%) reported both telehealth and oncology services. 272 hospitals (6.0%) offered oncology without telehealth, 1,369 (30.2%) offered telehealth without oncology, and 845 (18.6%) hospitals offered neither. 1,288 out of 3,152 counties with 26.6 million residents across 41 states had no hospital-based access to either oncology or telehealth. After adjustment, rural hospitals were less likely than urban hospitals to offer telehealth alongside existing oncology care (OR 0.27; 95% CI 0.14–0.55; p < .001). No county-level factors were significantly associated with telehealth availability among hospitals with oncology.

## Conclusions and relevance

Hospital-based cancer care and telehealth are widely available across the US; however, 8.4% of patients are at risk for geographic barriers to cancer care. Advocacy for adoption of telehealth is critical to ensuring equitable access to high-quality cancer care, ultimately reducing place-based outcomes disparities. Detailed, prospective, data collection on telehealth utilization for cancer care is also needed to ensure improvement in geographic access inequities.

## Introduction

Cancer patients in rural areas of the United States experience worse outcomes across all phases of cancer care in comparison to non-rural populations, in part due to geographic barriers to accessing high-quality oncologic care. For example, geographic barriers to care have been associated with decreased likelihood of high-quality treatment for patients with breast [1], colon [2], rectal [3], esophagogastric [4], gynecologic [5], and lung cancers [6]. One challenge in the pursuit of equity in cancer care for rural patients is ensuring appropriate geographic distribution of access points to high-quality cancer care. Prior nationwide analyses have suggested a mismatch between the distribution of oncology care professionals and the patient populations in need of care [7–9]. This mismatch is further exacerbated by contemporary trends toward the consolidation of the most complex cancer care in high-volume centers, in part owing to the improvement in outcomes accompanying treatment in high-volume settings [10]. Rural cancer patients may therefore face substantial travel burdens to reach facilities capable of providing high-quality cancer care.

Telehealth is critical to reducing cancer care disparities across the rural-urban continuum [11]. Telehealth offers referral centers the ability to provide outreach into rural and otherwise underserved areas whose residents may have difficulty traveling for care. For community cancer practices, telehealth likewise offers the potential for expanded geographic outreach, but also allows oncologists to connect with other cancer specialists to extend to patients some of the benefits of subspecialty cancer care while undergoing treatment locally [12].

Little is currently known about the capacity of hospitals within the United States to utilize telehealth as a component of cancer care. Moreover, few data exist on the availability of hospital-based telehealth services prior to their rapid adoption during the SARS-CoV-2 pandemic beginning in 2020. We therefore conducted a national analysis of hospitals' pre-pandemic provision of telehealth and oncology services, along with geographic and sociodemographic correlates of access, and identify opportunities to improve utilization of this critical element of high-quality cancer care.

## Materials and methods

### Data sources

This study employed data from three sources: 1) 2019 American Hospital Association (AHA) Annual Hospital Survey and IT Supplement, 2) 2018 Area Health Resources Files from the Health Resources and Services Administration (HRSA), and 3) 2013 Urban Influence Codes (UIC) from the United States Department of Agriculture. The AHA survey is an annual survey administered to hospitals and collects information on hospital characteristics, including telehealth adoption, oncology services, hospital ownership, number of staffed beds, system affiliation, accreditation status by Joint Commission or DVN, teaching status, Commission on Cancer accreditation status, and payer mix for Medicare and Medicaid, as well as hospital geographic coordinates. The response rate on the 2019 AHA survey was 75.1%. Data from hospitals that did not respond to telehealth questions were imputed using both 2019 AHA IT Supplement and the historical data from 2015–2018.

Additionally, county-level characteristics including Primary Care Health Professional Shortage Area (HPSA), percent of in-county residents by age group (0–17, 18–39, 40–64, and 65 or older), by gender (female vs. male), by race/ethnicity (non-Hispanic White, non-Hispanic Black/African American, American Indian and Alaska Natives, Asian, Hispanic, Other Races), by poverty level status (100% federal poverty level), by insurance coverage status, by English proficiency, dissimilarity index between non-White and White residents in a county, and percent of households having broadband access were obtained from the 2018 Area Health Resources Files from HRSA [13]. The final analytic data set included 4,540 hospitals located in 3,152 counties across 50 states and Washington, D.C. To establish rurality categories, hospital counties were placed into three categories based upon UICs, a United States Department of Agriculture measure that categorizes counties based upon their population size and adjacency to metropolitan areas: urban (UIC class 1 or 2), rural micropolitan (UIC class 3, 5, or 8), and rural noncore (all other UIC classes).

### Measurement

This cross-sectional study has two primary outcomes: availability of hospital-level telehealth, and availability of hospital-level oncology services. First, we derived hospital-level telehealth availability from the AHA survey, based on each hospital's responses on whether a hospital owned or provided telehealth services for consultation and office visits in its hospital settings. Second, we identified availability of hospital-level oncology servicesaccording to survey responses on whether a hospital provided on-site "inpatient and outpatient services for patients with cancer, including comprehensive care, support and guidance in addition to patient education and prevention, chemotherapy, counseling and other treatment methods." Using these two dichotomous measures, we categorized hospitals into four groups (per their self-reported telehealth and oncology services provision): 1) neither telehealth nor oncology services, 2) telehealth but no oncology services, 3) oncology services but no telehealth, and 4) both telehealth and oncology services.

In order to identify geographic regions of the United States most amenable to intervention by county-level accessibility to oncology and telehealth services, we subsequently aggregated hospital-level oncology and telehealth services and categorized counties into three groups: 1) no telehealth or oncology (low access): no hospitals within the county, or hospitals within the county with neither telehealth nor oncology services provided, 2) telehealth or oncology only (moderate access): having hospitals that had either telehealth only or oncology only, 3) both telehealth and oncology (high access): having hospitals that have both telehealth and oncology services.

## Statistical analyses

First, we plotted hospital spatial coordinates and used the "spatial join" tool in ArcGIS Pro to determine the highest level of services available within a hospital in a county. We developed a descriptive map showing the highest level of available services within a hospital in each county with point locations of each hospital. Next, we identified sociodemographic correlates of the populations of counties with low-, moderate-, or high levels of access to telehealth for cancer care. Pearson's Chi-square tests and two-group t tests were employed as appropriate to calculate the differences in county characteristics between high-access groups and each of the two less-access groups: moderate-access and low-access groups. We used population estimates from the 2019 U.S. Census Bureau American Community Survey to sum the population across counties by their access to telehealth and oncology services within hospitals.

To identify differences in hospital characteristics by oncology and telehealth services provision, we also analyzed hospital characteristics associated with the reported ability to provide oncology services, telehealth services, both services, or neither service, irrespective of geographic location. Pearson's Chi-square tests were used to compare hospital characteristics across these four groups.

To further examine the role of hospital characteristics in oncology and telehealth services provisions, we constructed two separate generalized logistic mixed effects models treating county-level variables as random effects and hospital-level variables as fixed effects with a random intercept for each county. One model was conditional on a hospital with oncology services to examine the incremental likelihood of having telehealth when a hospital had oncology; the other model was conditional on a hospital with telehealth to examine the likelihood of oncology when a hospital had telehealth. The final hospital-level characteristics include hospital beds (<100, 100–299, 300 or more beds), hospital ownership (public federal, public non-federal, private non-profit, private for-profit hospitals), system affiliation (yes, no), accreditation by Joint Commission or DVN (yes, no), teaching hospitals (yes, no), and the ratios of Medicare and Medicaid inpatient days to total inpatient days, whether rural health clinics co-located at a hospital, and Accountable Care Organization (ACO) status (currently leading an ACO, currently participating in an ACO, previously led or participated in an ACO, or never led or participated in an ACO). In addition to these hospital factors, we also controlled for the indicator of Commission on Cancer accreditation (yes, no) when examining the likelihood of having telehealth among hospitals with oncology.

The final county-level models included hospital location (urban, rural micropolitan, rural noncore areas), Primary Care HPSA status (none, part of the hospital county, whole county), percentages of in-county residents that were 65 years old or older, females, and by race/ethnicity (non-Hispanic white, non-Hispanic Black, American Indian and Alaska Natives, Asian, Hispanic, and other non-Hispanic races), non-White versus White residential segregation (dissimilarity index), percentages of residents not proficient in English, percentages of households with broadband access, and percentages of residents without health insurance, as well as state indicators. All analysis were produced using SAS, version 9.4, Strata, version 13.1, and Microsoft Excel, version 16.43. Institutional Review Board approval was not required for this study as it does not qualify as human subjects research.

# Results and discussion

## Hospital-level variations in telehealth and oncology services provision

Of the 4,540 hospitals analyzed as part of the AHA database, 2,054 (45.2%) reported offering both telehealth and oncology services. 272 hospitals (6.0%) offered oncology without available

telehealth, 1,369 (30.2%) offered telehealth without oncology services, and 845 (18.6%) hospitals offered neither service. 2,054 (88.3%) of hospitals providing cancer care also provided care by telehealth, whereas 272 hospitals offered cancer care but no telehealth. Hospitals offering both telehealth and oncology services were substantially more likely than hospitals without either of these services to be private, non-profit institutions (75.3% vs. 36.5%, respectively; p<0.001), to have capacity ≥300 beds (32.4% vs 3.3%, respectively; p<0.001), to be affiliated with a hospital system (77.6% vs 47.9%, respectively; p<0.001), and to identify as a teaching hospital (11.9% vs 0.6%, respectively; p<0.001). Hospitals without either oncology or telehealth services were more likely than hospitals with both services to be in counties designated in their entirety as HPSAs (25.4% vs 3.2%, respectively; p<0.001), and to derive the lowest proportion of inpatient reimbursement from Medicaid (40.0% vs 17.5%, respectively; p<0.001: Table 1).

## County-level variations in hospital-based telehealth and oncology services availability

Of 3,152 United States counties, 1,288 had no hospital-based access to either oncology or tele-health services. These low-access counties, with an approximate population of 26.6 million Americans (8.1% nationally), either had no hospital, or hospitals within their borders had nei-ther service. Low-access counties were present in all but 8 states, mostly located in the North-east (ME, NH, MA, CT, RI, NJ, DE, HI). 754 moderate-access counties, with an approximate population of 27.4 million (8.4% nationally), had access to either telehealth or oncology, but not both. 1,110 high-access counties, with an approximate population of 274.4 million (83.5% nationally), had at least one hospital with both oncology and telehealth services (Fig 1). High-access counties were more likely to be classified as urban than moderate- and low-access coun-ties (59.3% vs. 22.9% and 26.1%, respectively; p<0.001). Low-access counties were substan-tially more likely to be designated in their entirety as a HPSA than high-access counties (46.3% vs 5.8%, respectively, p<0.001). Likewise, the mean population per primary care physician in low-access counties was 3,447 (SD 2,996), compared to 2,714 (SD 1,940) in moderate-access and 1,706 (SD 1,175) in high-access counties, p<0.001. The distribution of race and ethnicity across high, moderate, and low access counties was notable for a proportionately higher Amer-ican Indian/Alaska Native population in low access counties compared to moderate- and high-risk counties (10.6% vs 6.0% vs 3.8%, respectively; p<0.001, Table 2).

After adjustment, rural hospitals were less likely than urban hospitals to offer telehealth ser-vices alongside cancer care (OR 0.27; 95% CI 0.14–0.55; p < .001; Table 3). Telehealth avail-ability in hospitals providing oncology care was associated with ≥300 hospital beds compared to <100 beds (OR 2.26; 95% CI 1.31–3.87; p = 0.002), private non-profit ownership vs. public hospitals (OR 1.67; 95% CI 1.10–2.53; p = 0.015), system affiliation (OR 1.61; 95% CI 1.11–2.33; p = 0.04), Commission on Cancer accreditation (OR 2.12; 95% CI 1.43–3.15; p<0.001), and hospital leadership of an ACO compared to never participating in an ACO (OR 2.52; 95% CI 1.58–4.02; p<0.001). Similarly, among hospitals with telehealth, the likelihoods of oncology services provisions were higher among larger hospitals (≥300 beds vs. <100 beds; OR 36.78; 95% CI 22.43–60.32; p < .001), accredited hospitals (OR 2.54; 95% CI 1.92–3.35; p < .001), hospitals with higher ratios of Medicare and Medicaid inpatient days, and those that are cur-rently leading an ACO (OR 1.44; 95% CI 1.07–1.92; p = 0.014). There were no significant county-level factors associated with telehealth availability among hospitals with oncology ser-vices, whereas county-level poverty rates and uninsured rates were associated with lower odds of having oncology services among hospitals that offered telehealth.

Telehealth is a critical component of strategies designed to improve health equity by reduc-ing rural-urban disparities in cancer outcomes. Our national study found that nearly half of

**Table 1. Hospital characteristics by telehealth and oncology services provision status, 2019.**

| | All Hospitals | Telehealth and Oncology Services Provision | | | | P values for the differences |
| --- | --- | --- | --- | --- | --- | --- |
| | | Neither Oncology Nor Telehealth | Telehealth but no Oncology | Oncology but no Telehealth | Both Oncology and Telehealth | |
| | | Number of Hospitals | | | | |
| **All** | 4,540 | 845 | 1,369 | 272 | 2,054 | |
| **Hospital Location** | N | Column Percent | | | | < .001 |
| Urban | 2,684 | 44.5 | 39.7 | 70.6 | 76.5 | |
| Rural Micropolitan | 754 | 15.3 | 21.0 | 16.2 | 14.3 | |
| Rural Noncore | 1,102 | 40.2 | 39.2 | 13.2 | 9.2 | |
| **Hospital Ownership** | | | | | | < .001 |
| Public non-federal | 952 | 31.8 | 26.7 | 19.9 | 12.8 | |
| Private non-profit | 2,745 | 36.5 | 53.5 | 58.1 | 75.3 | |
| Private for-profit | 651 | 21.9 | 16.2 | 16.2 | 9.7 | |
| Public federal | 192 | 9.8 | 3.5 | 5.9 | 2.2 | |
| **Hospital beds** | | | | | | < .001 |
| <100 | 2,306 | 76.1 | 77.7 | 32.0 | 24.9 | |
| 100–299 | 1,442 | 20.6 | 19.3 | 47.1 | 42.7 | |
| 300+ | 792 | 3.3 | 3.0 | 21.0 | 32.4 | |
| **System Affiliation** | | | | | | < .001 |
| Yes | 3,069 | 47.9 | 64.7 | 68.0 | 77.6 | |
| No | 1,471 | 52.1 | 35.3 | 32.0 | 22.4 | |
| **Accreditation by Joint Commission or DVN** | | | | | | < .001 |
| Yes | 3,331 | 54.6 | 57.6 | 87.1 | 89.8 | |
| No | 1,209 | 45.4 | 42.4 | 12.9 | 10.2 | |
| **Teaching Hospital** | | | | | | < .001 |
| Yes | 270 | 0.6 | 0.5 | 4.8 | 11.9 | |
| No | 4,270 | 99.4 | 99.5 | 95.2 | 88.1 | |
| **Commission on Cancer Accredited** | | | | | | < .001 |
| Yes | 1,270 | - | - | 51.5 | 55.0 | |
| No | 3,270 | 100.0 | 100.0 | 48.5 | 45.0 | |
| **Ratio of Medicare Inpatient Days to Total Inpatient Days** | | | | | | < .001 |
| <45% | 1,190 | 27.6 | 26.0 | 23.9 | 26.1 | |
| 45%-55% | 1,379 | 34.6 | 24.3 | 43.4 | 31.0 | |
| >55% | 1,971 | 37.9 | 49.7 | 32.7 | 42.9 | |
| **Ratio of Medicaid Inpatient Days to Total Inpatient Days** | | | | | | < .001 |
| Tertile I: <10% | 1,273 | 40.0 | 38.0 | 20.2 | 17.5 | |
| Tertile II: 10–20% | 1,438 | 25.9 | 28.4 | 23.2 | 37.3 | |
| Tertile III: >20% | 1,829 | 34.1 | 33.6 | 56.6 | 45.1 | |
| **Health Professional Shortage Area—Primary Care, 2020** | | | | | | < .001 |
| None | 367 | 6.5 | 6.9 | 8.8 | 9.4 | |
| Part county | 3,610 | 68.1 | 73.0 | 88.6 | 87.4 | |
| Whole county | 563 | 25.4 | 20.2 | 2.6 | 3.2 | |
| **Rural Health Clinics Co-located at Hospital** | | | | | | < .001 |
| Yes | 1,009 | 16.1 | 32.8 | 15.1 | 18.7 | |
| No | 2,380 | 18.0 | 40.3 | 53.3 | 74.6 | |

(*Continued*)

**Table 1.** (Continued)

| | All Hospitals | Telehealth and Oncology Services Provision | | | | P values for the differences |
| --- | --- | --- | --- | --- | --- | --- |
| | | Neither Oncology Nor Telehealth | Telehealth but no Oncology | Oncology but no Telehealth | Both Oncology and Telehealth | |
| Missing | 1,150 | 65.9 | 27.0 | 31.6 | 6.8 | |
| **Accountable Care Organization (ACO)** | | | | | | < .001 |
| Hospital/system current leads an ACO | 972 | 1.8 | 14.2 | 11.8 | 35.6 | |
| Hospital/system currently participates in an ACO (but not its leader) | 747 | 4.5 | 17.5 | 16.5 | 20.7 | |
| Hospital/system previously led or participated in an ACO | 123 | 1.0 | 2.3 | 1.8 | 3.9 | |
| Hospital/system has never participated or led an ACO | 2,697 | 92.8 | 66.1 | 69.9 | 39.9 | |

Notes: Pearson's Chi-square tests were employed to calculate the differences in hospital characteristics by oncology and telehealth services provision.

hospitals have both oncology and telehealth capabilities, and an additional 30% of hospitals had telehealth without oncology care capacity. Hospitals may utilize telehealth to deliver cancer treatment, monitor for toxicities, offer survivorship care, or provide palliative and postoperative care when travel for an in-person visit is not feasible. However, we found that with

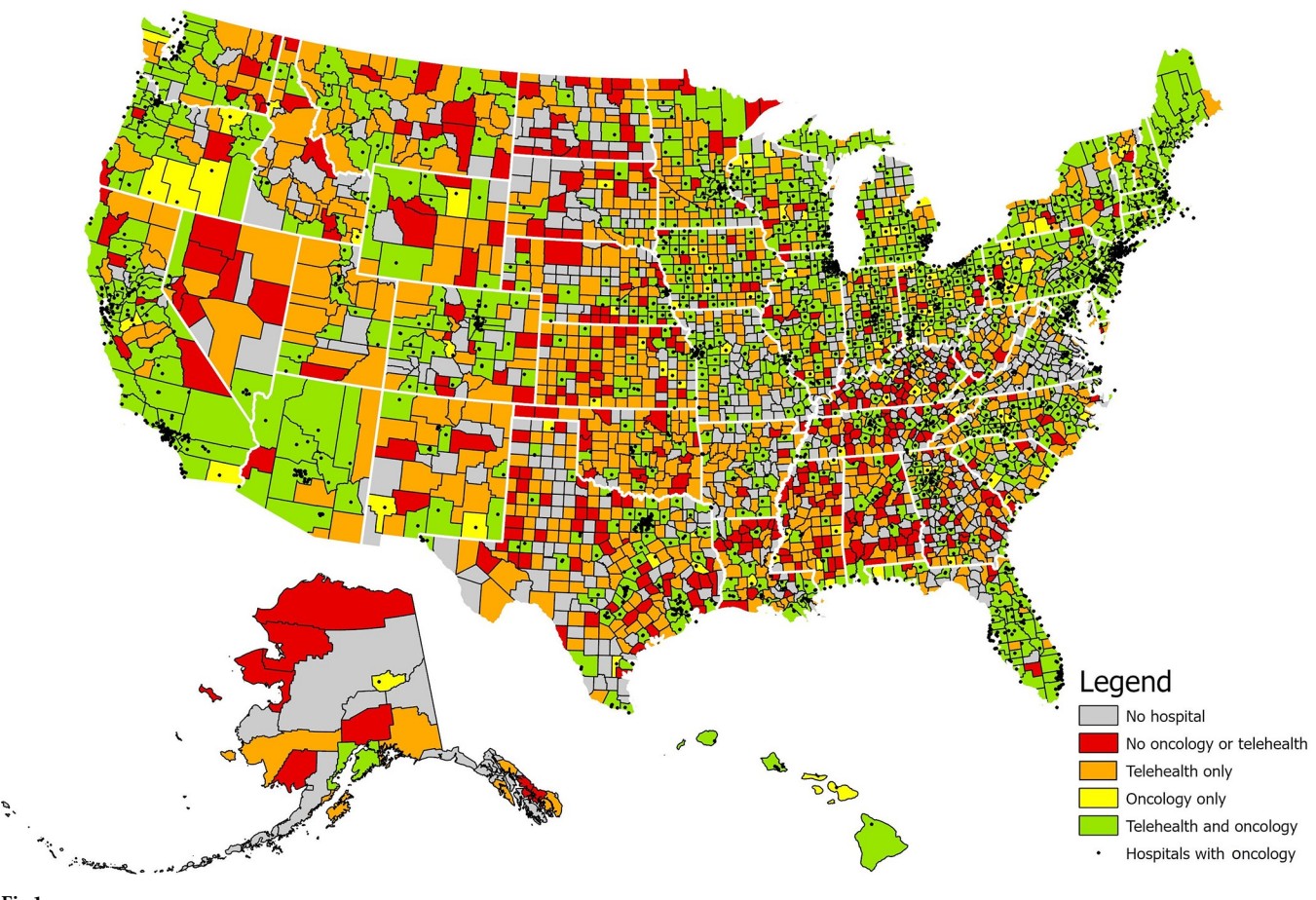

**Fig 1.**

**Table 2. County characteristics by telehealth and oncology services provision status, 2019.**

| | No Oncology or Telehealth (N = 1,288) | | Compared to Counties with Both | Telehealth or Oncology Only (n = 754) | | Compared to Counties with Both | Both Oncology and Telehealth (n = 1,110) | |
|---|---|---|---|---|---|---|---|---|
| | N | Col % | P | N | Col % | P | N | Col % |
| **County Rurality†** | | | < .001 | | | < .001 | | |
| Urban (n = 1,167) | 336 | 26.1% | | 173 | 22.9% | | 658 | 59.3% |
| Rural Micropolitan (n = 650) | 188 | 14.6% | | 190 | 25.2% | | 272 | 24.5% |
| Rural Noncore (n = 1,335) | 764 | 59.3% | | 391 | 51.9% | | 180 | 16.2% |
| **Health Professional Shortage Area—Primary Care, 2020** | | | < .001 | | | < .001 | | |
| None (n = 334) | 129 | 10.0% | | 68 | 9.0% | | 137 | 12.3% |
| Part county (n = 1,946) | 563 | 43.7% | | 474 | 62.9% | | 909 | 81.9% |
| Whole county (n = 872) | 596 | 46.3% | | 212 | 28.1% | | 64 | 5.8% |
| | **Mean** | **Std** | | **Mean** | **Std** | | **Mean** | **Std** |
| **% in-County Residents by Age, 2018** | | | | | | | | |
| Age 0–17 | 21.3% | 3.9% | 0.208 | 22.1% | 3.2% | < .001 | 21.5% | 2.9% |
| Age 18–39 | 23.6% | 3.4% | < .001 | 24.3% | 3.1% | < .001 | 26.6% | 4.0% |
| Age 40–64 | 31.8% | 3.3% | 0.004 | 31.2% | 2.7% | 0.043 | 31.5% | 2.8% |
| Age 65+ | 23.3% | 6.4% | < .001 | 22.4% | 5.1% | < .001 | 20.5% | 4.8% |
| **% In-County Residents that are Females, 2018** | 49.4% | 2.8% | < .001 | 49.7% | 2.0% | < .001 | 50.5% | 1.5% |
| **% In-County Residents by Race/Ethnicity, 2018** | | | | | | | | |
| Non-Hispanic White | 76.0% | 21.7% | 0.474 | 76.9% | 19.9% | 0.090 | 75.4% | 18.6% |
| Non-Hispanic Black | 9.9% | 16.2% | 0.110 | 7.5% | 13.6% | 0.016 | 9.0% | 12.3% |
| American Indian and Alaska Natives | 3.2% | 10.6% | < .001 | 2.3% | 6.0% | < .001 | 1.4% | 3.8% |
| Asian | 0.9% | 2.1% | < .001 | 1.0% | 1.6% | < .001 | 2.7% | 4.0% |
| Hispanic | 8.7% | 14.1% | 0.030 | 10.9% | 15.8% | 0.100 | 9.9% | 11.9% |
| Other Races | 2.1% | 2.1% | < .001 | 2.2% | 1.9% | 0.2% | 2.5% | 2.2% |
| **% Residents in Poverty, 2018** | 16.5% | 6.9% | < .001 | 15.5% | 5.6% | < .001 | 13.4% | 5.0% |
| **Median Household Income, 2018** | $ 49,365 | $ 12,368 | < .001 | $ 50,094 | $ 10,577 | < .001 | $ 58,573 | $ 15,543 |
| **Non-White/White Residential Segregation, 2015–2019** | 27.8% | 14.0% | < .001 | 29.2% | 12.8% | < .001 | 34.9% | 11.4% |
| **% not Proficient in English, 2015–2019** | 1.4% | 2.9% | < .001 | 1.7% | 2.7% | 0.035 | 2.0% | 2.7% |
| **% with Broadband Access, 2015–2019** | 71.9% | 9.1% | < .001 | 74.0% | 8.1% | < .001 | 80.4% | 6.5% |
| **% In-County Residents Uninsured, 2018** | 12.5% | 5.1% | < .001 | 12.4% | 5.2% | < .001 | 9.7% | 4.4% |
| **% In-County Residents Covered by Medicare, 2018** | 24.6% | 5.8% | < .001 | 23.9% | 5.0% | < .001 | 21.9% | 5.1% |
| **Ratio of population to primary care physicians, 2018** | 3,447 | 2,996 | < .001 | 2,714 | 1,940 | < .001 | 1,706 | 1,175 |

Notes: Pearson's Chi-square tests and two-group t tests were employed as appropriate to calculate the differences in county characteristics by oncology and telehealth services provision† Hospital counties were categorized based on 12-group Urban Influence Codes (UIC) into three locations: Urban (UIC class 1 or 2), rural micropolitan (UIC class 3, 5, or 8), and rural noncore (all other UIC classes).

increasing rurality or freestanding status, there is an increased likelihood of lacking telehealth within a county or hospital, and that tens of millions of Americans live in counties without either telehealth or oncology services available at hospitals. As hospitals in counties with lower income and higher uninsured rates were less likely to have oncology care alongside existing telehealth capacity, expansion of oncology services to these hospitals may help address cancer

**Table 3. Factors associated with availability of telehealth care in hospitals providing oncology services in 2019.**

| | Among Hospitals with Oncology Care | Among Hospitals with Telehealth |
| --- | --- | --- |
| | Telehealth vs. No Telehealth | Oncology vs. No Oncology |
| **Hospital Location** | | |
| Urban | ref. | ref. |
| Rural Micropolitan | **0.36 (0.21, 0.65)**\*** | 0.80 (0.57, 1.11) |
| Rural Noncore | **0.27 (0.14, 0.55)**\*** | **0.43 (0.30, 0.63)**\*** |
| **Hospital Beds** | | |
| <100 | ref. | ref. |
| 100–299 | 1.50 (0.99, 2.29) | **6.66 (4.96, 8.94)**\*** |
| 300+ | **2.26 (1.31, 3.87)**\** | **36.78 (22.43, 60.32)**\*** |
| **Hospital Ownership** | | |
| Public | ref. | ref. |
| Private non-profit | **1.67 (1.10, 2.53)**\* | 1.12 (0.84, 1.49) |
| Private for-profit | 1.63 (0.92, 2.88) | **0.65 (0.44, 0.96)**\* |
| **System Affiliation** | | |
| Yes vs. No | **1.61 (1.11, 2.33)**\* | **0.74 (0.57, 0.96)**\* |
| **Accreditation by Joint Commission or DVN** | | |
| Yes vs. No | 1.11 (0.67, 1.86) | **2.54 (1.92, 3.35)**\*** |
| **Commission on Cancer Accredited** | | |
| Yes vs. No | **2.12 (1.43, 3.15)**\*** | - |
| **Ratio of Medicare Inpatient Days to Total Inpatient Days** | | |
| <45% | ref. | ref. |
| 45%-55% | 0.69 (0.45, 1.06) | **1.77 (1.31, 2.38)**\*** |
| >55% | 0.72 (0.45, 1.15) | **1.49 (1.11, 2.00)**\** |
| **Ratio of Medicaid Inpatient Days to Total Inpatient Days** | | |
| Tertile I: <10% | ref. | ref. |
| Tertile II: 10–20% | 1.12 (0.71, 1.76) | **1.90 (1.45, 2.49)**\*** |
| Tertile III: >20% | 0.82 (0.52, 1.29) | **1.41 (1.05, 1.89)**\* |
| **Health Professional Shortage Area—Primary Care, 2020** | | |
| None | ref. | ref. |
| Part county | 1.94 (0.67, 5.62) | **0.30 (0.18, 0.49)**\*** |
| Whole county | 1.05 (0.60, 1.85) | **0.67 (0.46, 0.98)**\* |
| **Rural Health Clinics Co-located at Hospital** | | |
| Yes | 1.56 (0.98, 2.49) | **1.35 (1.03, 1.76)**\* |
| No | ref. | ref. |
| Missing | **0.13 (0.08, 0.23)**\*** | **0.11 (0.08, 0.17)**\*** |
| **Accountable Care Organization (ACO)** | | |
| Hospital/system current leads an ACO | **2.52 (1.58, 4.02)**\*** | **1.44 (1.07, 1.92)**\* |
| Hospital/system currently participates in an ACO (but not its leader) | 1.25 (0.82, 1.90) | 1.18 (0.90, 1.56) |
| Hospital/system previously led or participated in an ACO | 1.97 (0.73, 5.32) | 1.48 (0.81, 2.71) |
| Hospital/system has never participated or led an ACO | ref. | ref. |
| **County Characteristics of Hospital Location** | | |
| % In-County Residents Age 65 or Older, 2018 | 1.09 (0.70, 1.70) | 1.18 (0.90, 1.55) |

*(Continued)*

**Table 3.** (Continued)

| | Among Hospitals with Oncology Care | Among Hospitals with Telehealth |
| --- | --- | --- |
| | Telehealth vs. No Telehealth | Oncology vs. No Oncology |
| % In-County Residents that are Females, 2018 | 1.17 (0.29, 4.67) | 1.79 (0.81, 3.98) |
| % In-County Residents by Race/Ethnicity, 2018 | | |
| Non-Hispanic White | | |
| Non-Hispanic Black | 0.94 (0.80, 1.11) | 1.00 (0.89, 1.12) |
| American Indian and Alaska Natives | 2.18 (0.79, 6.03) | 1.01 (0.78, 1.31) |
| Asian | 0.98 (0.63, 1.51) | 0.99 (0.67, 1.44) |
| Hispanic | 0.82 (0.65, 1.03) | 0.95 (0.81, 1.13) |
| Other Races | 0.66 (0.32, 1.35) | 0.85 (0.47, 1.53) |
| % Residents in Poverty, 2018 | 0.79 (0.46, 1.36) | **0.70 (0.50, 0.97)**[*] |
| Non-White/White Residential Segregation, 2015–2019 | 1.03 (0.88, 1.22) | **1.19 (1.08, 1.31)**[***] |
| % not Proficient in English, 2015–2019 | 1.02 (0.38, 2.69) | 1.07 (0.50, 2.30) |
| % with Broadband Access, 2015–2019 | 1.02 (0.66, 1.57) | **1.49 (1.17, 1.89)**[**] |
| % In-County Residents Uninsured, 2018 | 1.22 (0.77, 1.94) | **0.61 (0.45, 0.83)**[**] |

Notes: Two separate generalized logistic mixed-effects regressions were conducted with random effects for hospital counties.

care access deficiencies. Although data are lacking on sustained, post-pandemic implementation of telehealth for cancer care, it is likely that access disparities continue for rural populations.

It is crucial to encourage hospitals caring for cancer patients without the availability of telehealth to sustainably adopt this critical adjunct to cancer care, if they have not already done so post-pandemic. Telehealth for cancer care might include virtual clinic visits, virtual supervision of therapy, remote patient monitoring, or clinician-to-clinician consultation [14]. Our analysis indicates that hospitals caring for cancer patients without the benefits of telehealth are more likely to be rural or micropolitan, and not affiliated with a hospital system. These findings suggest that rural hospitals might benefit from advocacy for connection via telehealth to referral centers so that their patients have the benefit of subspecialty consultation when needed, without burdensome long-distance travel for care. Importantly, connections between referral centers and rural hospitals requires commitments to outreach care from both practice settings to ensure patients have adequate access to cancer telehealth. While both patient-to-clinician and clinician-to-clinician applications of telehealth have the potential to improve rural cancer care, clinician-to-clinician telemedical communication may have the highest yield for integration of oncologic referral centers with outlying community cancer centers. Specific applications warranting further investigation include remote "curbside" consultation, telementoring, multidisciplinary tumor boards, and care coordination.

Our analysis identifies geographic areas of the United States whose residents are at elevated risk for barriers to high-quality cancer care. 26.6 million people in 41 states reside within low-access counties without any hospital-based access to cancer care or telemedical infrastructure that might connect them to oncologists. It is critical for advocates for health equity at the state and federal levels to ensure that mechanisms are in place to improve local care for these counties' residents or ensure that they are able to access resources in nearby counties. Even if counties do not have ready access to oncology care, telehealth may facilitate tele-oncology services, virtual tumor boards and the like to help ensure that patients receive quality cancer diagnosis

and treatment [11]. The loosening of Medicare telehealth regulations and promotion of payment parity during the SARS-CoV-2 pandemic, the potential for those regulations to be extended beyond the pandemic, the flow of pandemic-related telehealth funding, and teleoncology innovations may help incentivize expansion of telehealth services in the future [15,16]. Importantly, telehealth expansion should occur in the context of broader efforts to improve digital health equity, including increasing availability of broadband internet access for rural populations.

Our findings found specifically that American Indian/Alaska Native populations have less access to telemedicine for cancer care, consistent with earlier work on this populations' barriers to high-quality oncologic services [7,8,17]. While rural populations have higher cancer-related mortality rates than urban patients, among rural populations, American Indian/Alaska Native populations have the highest rural-urban disparity in cancer mortality [18,19]. Given the added complexities of the Indian Health Service system and related requirements for tribal affiliation and/or residence on reservation lands, American Indian/Alaska Native residents may be particularly burdened to find culturally competent and low-cost care locally [20].

Hospitals that report offering both telehealth and oncology services may not currently be offering telehealth specifically for cancer care. These institutions should strongly consider including oncology in their telehealth services; we anticipate that the investment required for expansion of telehealth to cancer care is likely less than that required for health systems to adopt telehealth *de novo*. Additionally, available hospital-based data currently do not distinguish between clinician-patient applications of telehealth (e.g., virtual clinic visits and remote patient monitoring) and clinician-clinician applications (e.g., remote "curbside" consultations, or multidisciplinary cancer treatment conferences). These limitations of available data suggest an opportunity to improve datasets, like those generated by the AHA survey, to capture detailed information on specific telehealth applications to guide expansion of access to remote cancer care and other service lines amenable to telehealth.

Our analysis is limited in several key aspects, related to the data available through the AHA. First, our focus on hospitals who provide outpatient and/or consultative care via telehealth may exclude hospitals that provide telehealth only in the context of eICU, inpatient, or addiction services; however, we believe that these latter use cases are somewhat less relevant to access to cancer care. Second, hospitals' reported use of telehealth service does not specify whether the hospital is the originator or acceptor of telehealth consultations (or both). To the extent that there might be telemedical communication among hospitals, we are therefore unable to determine which hospitals function as "hubs" or "spokes" for outreach care. Third, some rural hospitals may have responded to the AHA survey jointly with larger parent hospitals within their health system, thereby overestimating the resources available in rural hospitals. Fourth, while arguably representing the best data available, the AHA respondent set does not represent the entirety of cancer care delivery in the United States. While we imputed data for 2019 non-responders to the AHA survey from prior years, not all cancer care locations are categorized as hospitals. Finally, the 2019 AHA survey does not capture changes in the utilization of telehealth services attributable to the SARS-CoV-2 pandemic beginning in early 2020. Importantly, however, our analysis both represents the most contemporary data available on hospital-based telehealth infrastructure, and serves as a necessary pre-pandemic baseline for future investigations into this important topic. Overall telehealth utilization appears to have peaked in April, 2020 with claims decreasing by approximately half soon thereafter, and continuing to decrease in early 2022 [21,22]. Additionally, in early 2021, oncology was estimated to account for only 8% of all telehealth claims, suggesting that even if hospital-based infrastructure increased during the pandemic, utilization remains low [22].

## Conclusions

While hospital-based cancer care and telehealth resources are widely available across the United States, a substantial minority of patients across 42 states as well as American Indian/ Alaska Native residents are at risk for geographic barriers to high-quality cancer care. We recommend continuing to monitor the impact of SARS-CoV-2 pandemic-related policies and innovations on the availability of telehealth services for oncology care. Furthermore, improvements in data collection, and advocacy to prioritize adoption of telehealth as a critical element of cancer care, will improve health equity by decreasing disparities attributable to place of residence.

## Author Contributions

**Conceptualization:** David I. Shalowitz, Whitney E. Zahnd, Jan Eberth.

**Data curation:** David I. Shalowitz, Peiyin Hung, Whitney E. Zahnd.

**Formal analysis:** Whitney E. Zahnd.

**Investigation:** David I. Shalowitz, Peiyin Hung, Whitney E. Zahnd, Jan Eberth.

**Methodology:** David I. Shalowitz.

**Project administration:** David I. Shalowitz, Jan Eberth.

**Resources:** David I. Shalowitz.

**Supervision:** David I. Shalowitz, Jan Eberth.

**Visualization:** Whitney E. Zahnd.

**Writing – original draft:** David I. Shalowitz, Peiyin Hung, Whitney E. Zahnd.

**Writing – review & editing:** David I. Shalowitz, Peiyin Hung, Whitney E. Zahnd, Jan Eberth.

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
