## [Decision Letter · Decision Letter 0]

28 Nov 2022

PONE-D-22-22633Geographic Access to Hospital-Based Telehealth for Cancer CarePLOS ONE

Dear Dr. Shalowitz,

Thank you for submitting your manuscript to PLOS ONE. After careful consideration, we feel that it has merit but does not fully meet PLOS ONE’s publication criteria as it currently stands. Therefore, we invite you to submit a revised version of the manuscript that addresses the points raised during the review process.

I have now received reports from three reviewers. Based on the assessment and taking into consideration the feedback, I would like to invite you to revise your manuscript and provide a point-by-point rebuttal to all the comments.

We look forward to receiving your revised manuscript.

Kind regards,

Sonu Menachem Maimonides Bhaskar, MD PhD

Academic Editor

PLOS ONE

Journal Requirements

2. Please ensure that you have specified (1) whether consent was informed and (2) what type you obtained (for instance, written or verbal, and if verbal, how it was documented and witnessed). If your study included minors, state whether you obtained consent from parents or guardians. If the need for consent was waived by the ethics committee, please include this information.

Dr. Shalowitz received funding via the Wake Forest Comprehensive Cancer Center NCI cancer support grant (P30CA012197), and has received consulting fees from Nimble Co, LLC (purview.net) unrelated to the current work.

Dr. Shalowitz received funding via the Wake Forest Comprehensive Cancer Center NCI cancer support grant (P30CA012197), and has received consulting fees from Nimble Co, LLC unrelated to the current work. 

7. Please include a caption for figure 1.

8. We note that Figure 1 in your submission contain [map/satellite] images which may be copyrighted. All PLOS content is published under the Creative Commons Attribution License (CC BY 4.0), which means that the manuscript, images, and Supporting Information files will be freely available online, and any third party is permitted to access, download, copy, distribute, and use these materials in any way, even commercially, with proper attribution. For these reasons, we cannot publish previously copyrighted maps or satellite images created using proprietary data, such as Google software (Google Maps, Street View, and Earth). For more information, see our copyright guidelines: http://journals.plos.org/plosone/s/licenses-and-copyright.

9. Please include your tables as part of your main manuscript and remove the individual files. Please note that supplementary tables (should remain/ be uploaded) as separate "supporting information" files

Additional Editor Comments:

I have now received reports from three reviewers. Based on the assessment and taking into consideration the feedback, I would like to invite you to revise your manuscript and provide a point-by-point rebuttal to all the comments.

Reviewers' comments:

Reviewer's Responses to Questions

**Comments to the Author**

1. Is the manuscript technically sound, and do the data support the conclusions?

Reviewer #1: Yes

Reviewer #2: Yes

Reviewer #3: Partly

2. Has the statistical analysis been performed appropriately and rigorously? 

Reviewer #1: Yes

Reviewer #2: Yes

Reviewer #3: Yes

3. Have the authors made all data underlying the findings in their manuscript fully available?

Reviewer #1: Yes

Reviewer #2: Yes

Reviewer #3: Yes

4. Is the manuscript presented in an intelligible fashion and written in standard English?

Reviewer #1: Yes

Reviewer #2: Yes

Reviewer #3: Yes

5. Review Comments to the Author

Reviewer #1: This manuscript provides an interesting overview of the telehealth and oncology services in US hospitals prior to the pandemic. The authors are to be commended for the readable and logical organization of the manuscript. A few comments relating to the methods and certain key design decisions are noted below.

1. It is not clear if the availability of telehealth services correlate with the quality of the care provided at a given hospital. Could the authors comment on the quality of the services provided by these hospitals?

2. Do the hospitals that provide telehealth services, use telehealth to provide oncology care? Are there data to show what the telehealth services in these hospitals were used for in the pre-pandemic era?

3. It is difficult to assess policy/clinical implications of this work without knowing how the lack of availability of services/care relate to cancer outcomes--how do the differences in availability of services translate to disparities in cancer outcomes?

4. The results show that larger hospitals are 37 times more likely to provide oncology services among those hospitals that provide telehealth. Most of the smaller/rural hospitals are less likely to provide oncology care or access to telehealth services—perhaps these hospitals are more likely to refer cancer patients to nearby larger hospitals for cancer care? If that is true, the availability of telehealth services in larger hospitals that provide oncology services could perhaps help reduce cancer care deficiencies.

Overall, it is not clear if the greatest improvement in health outcomes depend on increasing oncology and telehealth services in rural hospitals OR further increasing the telehealth capacity in existing larger hospitals. It is not clear how a policy-maker could use this study to improve care for the population.

Reviewer #2: Reviewer name: Alma Nurtazina

Geographic Access to Hospital-Based Telehealth for Cancer Care

Corresponding author: David Shalowitz, M.D.. MSHP

Wake Forest University School of Medicine

Winston-Salem, NC UNITED STATES

The authors present the results of a cross-sectional study based on the data from three main surveys conducted in 2013-2019 in the USA. In total, data from 4540 hospitals were analyzed for telehealth and oncology services.

The title is expected to contain a place of study because of the high variability of telehealth services across different countries. The abstract describes the essential information in the work. The paper’s goal is within the scope of the journal. The introductory section adequately explains the framework of the research. The importance of the research idea is evident. The authors have applied an appropriate methodology to their research and clearly described it. The presentation of the study is complete for a scientific paper. The results are described and analyzed adequately. The article is written clearly and logically consistently. The conclusion is logically supported by the obtained results.

There are a few minor limitations:

1. The title does not contain the place of the study.

2. The authors did not mention study design which is a cross-sectional.

3. There are several errors in the tables’ design and data presentation.

4. The reference list is incomplete.

Reviewer #3: Thank you for giving me the opportunity to review this research, which evaluates US hospitals’ ability to provide equitable access to oncology services via telehealth. This is an important study that highlights the number of Americans with cancer who had to attend all cancer appointments in person prior to the pandemic. This is a well-written manuscript. However, considering that the data for telehealth utilization are pre-pandemic (and therefore of limited utility to today’s researchers and policymakers), the authors need to ensure that they are appropriately contextualizing the research to highlight its relevance. Please see below for specific areas for improvement.

Major

1. Thank you for acknowledging that these data are pre-pandemic and therefore likely do not represent the current use of telemedicine in hospitals. This is a significant limitation given what we know about the uptake of telemedicine in the US since 2020, and should be reflected in the title. Further, in order to make this relevant to today’s researchers and policy makers, the authors should consider bringing this limitation to the forefront of the discussion-- currently it is not mentioned until late in the discussion. For example, in the first discussion paragraph, when you discuss increasing rurality being associated with lower telehealth rates, I suggest you discuss how this disparity is likely still the case today (some data should be available to support this), even though your analysis is pre-COVID.

2. In the second discussion paragraph, the authors write about how it is “crucial to encourage hospitals caring for cancer patients” to adopt telehealth. Again, this needs to be framed in the understanding that this may have already happened. Further, there should be more specifics included. For example, what do the authors mean by “encourage?” Through national policy? Medicare reform? There is a robust discussion currently around making certain telemedicine reimbursement waivers permanent beyond the Public Health Emergency, and this might be a good place to bring in some of these arguments to describe how hospitals would be encouraged to adopt these technologies.

3. The authors point out that only 8% of telehealth claims were for oncology in 2021. This statement needs to be contextualized in comparison to other specialties, and regarding what percent is appropriate for oncology. For example, my understanding is that an initial visit for a cancer diagnosis needs to be done with an in person exam. Are there initial visits that are appropriate for telemedicine? Further, rural patients who are on active treatment typically have their visits stacked on the same day as infusion, radiation, or radiology visits (see: DeGuzman et al, (2015). Identifying barriers to navigation needs of cancer survivors in rural areas. Journal of Oncology Navigation & Survivorship, 6, 34-42), so a telehealth visit may not make sense in these cases.

4. It may be worthwhile to consider discussing that even as telehealth becomes more ubiquitous in hospitals, those living in rural areas often lack broadband, which raises its own set of challenges. Since COVID-19, alternative origination sites (such as public libraries), and the need to extend federal regulations that support reimbursement for telemedicine services.

Minor

1. In the section on Data Sources (page 4), it would be clearer to list the data sources in the same order that you explain them in that section (i.e. switch #2 and #3)

2. On page 5, line 133, it would be clearer to state what the outcomes are before explaining you they were derived: “Thus study has two primary outcomes: x and y.”

3. On page 5 lines 140-141, it would be clearer to put the statement “per their self-reported telehealth and oncology services provisions” in parenthesis

4. On page 8, line 199, although I was able to figure out where you got the number 2326 from in your table (hospitals providing cancer care), it took a while to figure it out and I wonder if this could be explained clearer.

6. PLOS authors have the option to publish the peer review history of their article (what does this mean?). If published, this will include your full peer review and any attached files.

Reviewer #1: No

Reviewer #2: **Yes: **Alma Nurtazina

Reviewer #3: No

---

## [Author Response · Author response to Decision Letter 0]

29 Dec 2022

See separately uploaded file labeled "Response to Reviewers."

---

## [Editor Report · Decision Letter 1]

16 Jan 2023

Pre-Pandemic Geographic Access to Hospital-Based Telehealth for Cancer Care in the United States

PONE-D-22-22633R1

Dear Dr. Shalowitz,

We’re pleased to inform you that your manuscript has been judged scientifically suitable for publication and will be formally accepted for publication once it meets all outstanding technical requirements.

Kind regards,

Sonu Bhaskar, MD PhD

Academic Editor

PLOS ONE

Additional Editor Comments (optional):

Thank you for submitting the revised version of your manuscript. I am pleased to accept the manuscript in its current form. Thank you for submitting your work to PLOS One.
---

## [Editor Report · Acceptance letter]

23 Jan 2023

PONE-D-22-22633R1 

Pre-Pandemic Geographic Access to Hospital-Based Telehealth for Cancer Care in the United States 

Dear Dr. Shalowitz:

I'm pleased to inform you that your manuscript has been deemed suitable for publication in PLOS ONE. Congratulations! Your manuscript is now with our production department. 

Kind regards, 

on behalf of

Dr. Sonu Bhaskar 

Academic Editor

PLOS ONE